# CdTiO$_3$-NPs incorporated TiO$_2$ nanostructure photocatalyst for scavenger-free water splitting under visible radiation

Nehal A. Erfan[1], Mohamed S. Mahmoud[1,2], Hak Yong Kim[3,4]*, Nasser A. M. Barakat[1]*

1 Chemical Engineering Department, Minia University, El-Minia, Egypt, 2 Collage of Applied Science, Department of Engineering, Suhar, Oman, 3 Department of Nano Convergence Engineering, Jeonbuk National University, Jeonju, South Korea, 4 Department of Organic Materials and Fiber Engineering, Jeonbuk National University, Jeonju, South Korea

* nasbarakat@mu.edu.eg (NAMB); khy@jbnu.ac.kr (HYK)

**Data Availability Statement:** The accepted submission contains the minimal data set; any researcher can get the required data and other

## Abstract

Nanofibrous morphology and the doping technique can overcome the problem of electron/hole fast recombination and improve the activity of titanium oxide-based photocatalysts. In this study, nanoparticulate and nanofibrous forms of CdTiO$_3$-incorporated TiO$_2$ were synthesized with different cadmium contents; the morphology and composition were determined by SEM, TEM, EDX, and XRD techniques. The nanomorphology, cadmium content, and reaction temperature of Cd-doped TiO$_2$ nanostructures were found to be strongly affect the hydrogen production rate. Nanofibrous morphology improves the rate of hydrogen evolution by around 10 folds over the rate for nanoparticles due to electron confinement in 0D nanostructures. The average rates of hydrogen production for samples of 0.5 wt.% Cd are 0.7 and 16.5 ml/g$_{cat}$.min for nanoparticles and nanofibers, respectively. On the other hand, cadmium doping resulted in increasing the hydrogen production rate from 9.6 to 19.7 ml/g$_{cat}$.min for pristine and Cd-doped (2 wt%) TiO$_2$ nanofibers, respectively. May be the formation of type I heterostructures between the TiO$_2$ matrix and CdTiO$_3$ nanoparticles is the main reason for the observed enhancement of photocatalytic activity due to the strong suppressing of electron/holes recombination process. Consequently, the proposed photocatalyst could be exploited to produce hydrogen from scavenger-free solution. Varying reaction temperature suggests that hydrogen evolution over the proposed catalyst is incompatible with the Arrhenius equation. In particular, reaction temperature was found to have a negative influence on photocatalytic activity. This work shows the prospects for using CdTiO$_3$ as a co-catalyst in photon-induced water splitting and indicates a substantial enhancement in the rate of hydrogen production upon using the proposed photocatalyst in nanofibrous morphology.

information from the included figures and tables in the manuscript.

**Funding:** The author(s) received no specific funding for this work.

**Competing interests:** The authors have declared that no competing interests exist.

## 1. Introduction

The depletion of fossil fuels has become a troubling fact. In addition to environmental restrictions, researchers are focusing on possible renewable energy-harvesting routes that can enable the world's energy demand to be continuously met. One such research path is to search for suitable energy carriers such as hydrogen and metals, which can be used to store renewable energy and employ it on demand. Several means of producing elemental hydrogen have been proposed, including control fermentation of waste biomasses [1], catalytic and thermal cracking of some hydrocarbons [2], the use of electrical energy to extract hydrogen from water [3], or the use of a photocatalyst for water splitting [4–6]. Photocatalytic water splitting is the best strategy due to its simplicity and its favorable economic factors. In addition, if the used photocatalyst works efficiently under visible light, the process can be considered a renewable energy technology [7, 8].

Among the proposed photocatalysts for water splitting, titanium dioxide has drawn the most attention due to its low cost and high chemical stability. With its band gap energy of 3.2 eV, TiO$_2$ is adequate for use under UV irradiation [9]. The high photogenerated electron/hole (e-/h)-pair recombination rate is a critical drawback of TiO$_2$, which strongly decreases its photocatalytic efficiency [10]. Such a fast recombination rate impedes the chemical reaction due to the short lifetime of the charge carriers (in the nanosecond range), meaning that there is not enough time for the photocatalytic reactions to take place [11]. Many attempts to solve these problems have been reported. The charge carriers' recombination at the grain boundaries can be strongly inhibited by controlling the internal crystalline structure to reduce the density of defects (such as TiO$_2$ [0 0 1] facets) [10]. In addition, non-metallic doping has attracted some research attention as a strategy for increasing the photo-excited electron's lifetime [12, 13]. For enhancing TiO$_2$ photocatalytic activity, many dopant materials such as carbonaceous materials (graphene, carbon nanotubes, fullerenes, and activated carbon) have been used [4, 6, 14–16].

Furthermore, coupling with other semiconductors was proposed in order to enhance photocatalytic activity by creating low-energy bands for the charge carriers. Typically, TiO$_2$ photocatalytic activity is enhanced through doping with transition metal nanoparticles. Excellent performance has been shown by such nanoparticles under UV and visible light irradiation [17–19]. Moreover, when the nanoparticles of metal oxides are electronically doped with aliovalent dopants, oxygen vacancies, or interstitial dopants, they exhibit the same behavior as metal nanoparticles due to the substantial concentration of charge carriers over the nanoparticle surface. Dopants in the photocatalyst act not only as recombination sites between photogenerated electrons and holes, but also as visible light absorption centers with an absorption coefficient that depends upon dopant density.

Moreover, a high Schottky barrier which enhance electron capture can be obtained by doping with foreign metal nanoparticles [20]. Electron capture results in an enlarged separation lifetime of the e$^-$/h pair. Increasing the e$^-$/h-pair separation lifetime decimates the recombination of these pairs and therefore enhances the transfer of holes and possibly electrons to O$_2$ adsorbed on the photocatalyst surface. Afterwards, the excited electrons are trapped by the foreign nanoparticles and the recombination of the e-/h pairs is suppressed. Furthermore, some researchers have indicated that doping TiO$_2$ with foreign nanoparticles can enhance rutile phase formation which has a greater tendency for visible light absorption than anatase [21, 22].

Besides the influence of the composition, the nanostructure morphology reveals a distinct effect upon the photocatalytic activity. It has been reported that the titania nanoparticles' quantum size of under 10 nm was the reason for the distinct improvement in photocatalytic

activity [23]. The photocatalytic electronic modification and the closeness of the excited e-/h pairs in nanostructures strongly contribute in the enhancement of the reaction which consequently improves their performances over those of larger titania powders. However, complicated and expensive processes are needed to synthesize TiO$_2$ quantum dots. Compared to nanoparticulate morphology, the large axial ratio of nanofibers strongly enhances the physicochemical and catalytic characteristics because of the rapid electron transfer, which markedly improves the activity of the TiO$_2$ photocatalyst [21, 24]. Among the several nanofiber synthesis techniques, electrospinning has drawn the most attention due to its low cost, simplicity, high yield, product morphology controllability, and applicability to a wide range of materials [25, 26].

Titanium-based perovskite-type oxides with formula of MTiO$_3$ (M = Cd, Ca, Ba, Sr, . . . etc.) show distinct ferroelectric, dielectric, piezoelectric, pyroelectric and photostrictive properties [27]. Among these materials, cadmium titanate (CdTiO$_3$) possesses interesting characteristics [28]. Accordingly, this semiconductor has been utilized in several applications including conductive material, optical fibers and humidity sensors [29–31]. Moreover, it has a strong potential in the photocatalysis applications [32]. CdTiO$_3$ crystallizes non-ferroelectric ilmenite or ferroelectric orthorhombic perovskite phases depending on the method of synthesis and/or post growth annealing temperature [33].

In this study, to investigate the influence of the nanomorphology and cadmium oxide doping, CdTiO$_3$-doped TiO$_2$ nanofibers were prepared by an electrospinning technique, and their photocatalytic activity for water splitting was evaluated. In addition, nanoparticulate photocatalysts with similar nanofiber compositions were prepared to study the effect of nanomorphology on the photocatalytic activity. Interesting results were recorded as the nanofibrous morphology strongly enhanced the photocatalytic activity. Besides the distinct enhancement in the hydrogen production rate, cadmium doping also improved the photocatalytic splitting of water to the point that it became incompatible with the Arrhenius equation—meaning that working under low temperature which is economically beneficial. Moreover, the proposed photocatalyst could be successfully exploited to produce hydrogen gas from a scavenger-free water.

## 2. Materials and methods

### 2.1 Materials

All chemicals in this study have been used without further modification. Titanium (IV) isopropoxide (TiIso, C$_{12}$H$_{28}$O$_4$Ti, 99.9 purity) and polyvinylpyrrolidone (PVP, (C$_6$H$_9$NO)$_n$ Mw = 130000) were purchased from Sigma Aldrich. Cadmium acetate di-hydrate (CdAc, Cd(CH$_3$CO$_2$)$_2$.2H$_2$O purity 99.0%) was obtained from Showa Co. Japan. Analytical-grade ethanol was used as a solvent (99.8–100% purity from Alpha chemicals). Acetic acid (99–100% purity), sodium sulfide (Na$_2$SO$_3$, Mw = 126, 98% purity), and sodium sulfate (Na$_2$SO$_4$, Mw = 78.04, 55–60% purity) were obtained from Alpha chemicals.

### 2.2 Fabrication of TiO$_2$ and Cd-doped TiO$_2$ nanofibers and nanoparticles

Fig 1 represents a schematic diagram for the nanofibers and nanoparticles fabrication procedure. For comparison, pure TiO$_2$ nanofibers were first prepared. A mixture of 2 g anhydrous ethanol and an equal amount of glacial acetic acid was prepared. Then, 1 g of Titanium isopropoxide was added to the solvent mixture. Later on, 1 g of PVP and 6 g of ethanol were added to the previous solution. The whole mixture was stirred at 300 rpm for 15 min to obtain a transparent sol–gel.

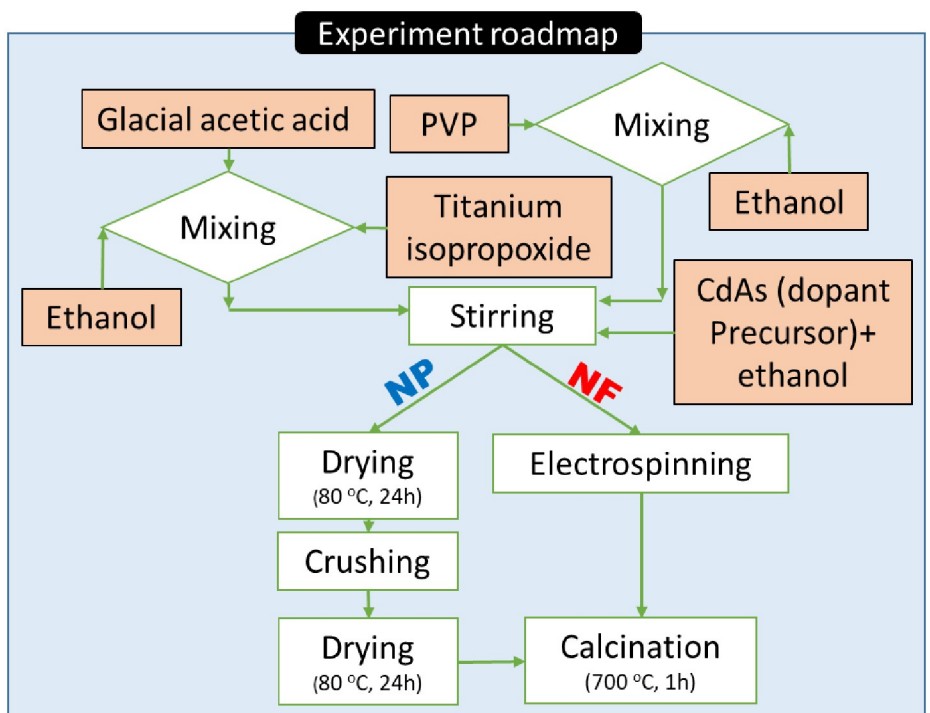

**Fig 1. Experimental procedure for Cd-doped TiO$_2$ nanofibers and nanoparticles preparation.**

Regarding the electrospinning experiment, the unit consists of a DC power supply, a spinning syringe mounted on a flexible syringe holder, and a rotating drum target. The distance from the syringe to the rotating drum was fixed at 15 cm. To perform a uniform deposition of the nanofiber, the rotation speed of the drum target is fixed as 10 rpm. High intensity DC voltage of 20 kV is applied between the syringe tip and the rotating drum. The drum was covered by a polyethylene sheet. After electrospinning, the mat underwent a drying step under vacuum at 80˚C for 24 h, followed by a calcination process at 700˚C for a 1-h holding time. In order to prepare Cd-doped TiO$_2$ nanofibers, the aforementioned procedure was similarly performed after the addition of specific amounts of CdAc dissolved in 1 ml ethanol. To prepare final nanofiber mats containing 0.5, 1, and 2 wt.% CdO with respect to titanium element, the corresponding mass of cadmium acetate is added. The calculations were based on assumptions that a complete elimination of the used polymer and full decompositions of titanium isopropoxide and cadmium acetate di-hydrate precursors into titanium oxide and cadmium oxide, respectively will carry out. The calcination process was done in air atmosphere. Moreover, to ensure performing complete oxidation reactions, suction pump was used to remove the gaseous products from the reaction atmosphere as well as to fill the tubular furnace by fresh air continuously. Accordingly, the claiming about complete elimination of the used polymer and converting the used metallic precursors to the oxide forms is acceptable. Considering that the melting points of titanium oxide and cadmium oxide are 1843 and 900 ˚C, respectively, no losses in the metal content are expected especially the calcination temperature was chosen to be 200 ˚C less than the melting point of the relatively volatile oxide (CdO). Using the same prepared solutions for nanofiber mats synthesis, nanoparticles with similar compositions were prepared by vacuously drying the solutions overnight, crushing and well grinding before calcination process.

## 2.3 Water photo-splitting experiment

The experiments were performed under a 1,000-W mercury lamp as a source of visible light. A solution containing 0.05 g of catalyst was added to 100 ml of 0.5 M Na$_2$S/Na$_2$SO$_3$ as a sacrificial agent. The general role of the scavenger agent in the water splitting process is to improve the hydrogen evolution. For Na$_2$S/Na$_2$SO$_3$ as a sacrificial agent, they capture of holes generated from the photocatalyst which leads to promoting the oxygen evolution reaction, in turn, the hydrogen reduction reaction is enhanced as well [34]. The suspension was placed in a well-sealed round-bottom flask with one opening from which a rubber tube exited. This tube was immersed in a water-filled inverted graduated cylinder; by displacing the water, the evolved gases were collected. In order to maintain the reaction temperature, the round-bottom flask was jacketed by a temperature-controlled water bath; the accumulated gas from the photocatalytic water-splitting reaction consists mainly of H$_2$ and O$_2$ with a molar ratio of 2:1. The number of moles of accumulated hydrogen was calculated by recording the change in the volume above the water level using the following equation:

$$n = \frac{2 \times 273 \times V}{3 \times 22.4 \times m \times T} \tag{1}$$

where $n$ is the number of moles of H$_2$ [mmol/g], $V$ is the volume of the gas (mL), $m$ is the mass of the photocatalyst (g), and $T$ is the temperature of the solution (K). In literature, the rate of hydrogen generation was represented different units including mmol/g$_{catalyst}$, μmol/h/g$_{catalyst}$, and ml/g$_{catalyst}$ according to the ideal gas law for hydrogen gas. However, for kinetic calculation, the rate of hydrogen evolution is expressed in mmol/min to calculate the quantum efficiency. It is noteworthy mentioning that to ensure the collected gas represents the water splitting reaction product gases and no interfere from water vapor especially the used lamp becomes hot during illuminating time, these precautions have been conducted. First, the lamp was cooling by a suction fan placed above the lamp so this used fan sucks the hot air around the reactor flask. Second, the collecting gas beaker was placed a way from the lamp so it was always cold. Third, the used water in the collector beaker was checked continuously during the reaction time to be sure it still cold. Fourth, after every experiment, the collect gas was left for relatively long time to allow any water vapor to condensate; no any change in the gas volume was observed which indicates the collected gas is free from any water vapor.

## 2.4 Characterization

The surface morphology of the as-obtained nanofibers and nanoparticles was studied using the JEOL JSM-5900 scanning electron microscope (JEOL Ltd., Japan). A Rigaku X-ray diffractometer (Rigaku Co., Japan) with Cu Kα (λ = 1.54056 Å) radiation over a 2θ range from 10˚ to 80˚ was used to characterize the phase and crystallinity of the prepared nanomaterials. A JEOL JEM-2200FS transmission electron microscope (TEM) operating at 200 kV and equipped with EDX (JEOL Ltd., Japan) was used to investigate the materials' internal structure.

# 3. Results and discussion

## 3.1 Catalyst characterization

During the calcination of the electrospun nanofibers in air, the utilized polymer is fully eliminated and the metallic precursor decomposes into its most stable form. Therefore, choosing a suitable precursor is a main reason to obtain good nanofiber morphology using the electrospinning technique. The main aspect of the proper precursor is polycondensation during sol–

gel preparation. The high polycondensation tendency and hydrolysis reactions for alkoxides explain their distinct performance in forming an integrated network [35–37].

The organometallic family comprises compounds having one or more metal atoms in the molecule, and metal alkoxides are members of that family. Metal alkoxides (R-O-M) were obtained by replacing the hydrogen atom in the hydroxyl group of alcohols (R-OH) with a metal atom M, which are the class of chemical precursors most widely used in sol–gel formation because of their condensation behavior and tendency to combine together to form a gel network. The gel network can be formed using metal salts such as chlorides, nitrates, and acetates besides alkoxides. The acetates showed the most convenient polycondensation behavior for gel network formation [38]. Excellent morphology and bead-free obtained nanofibers after the calcination process shown in Fig 2A indicates that the electrospinning working parameters

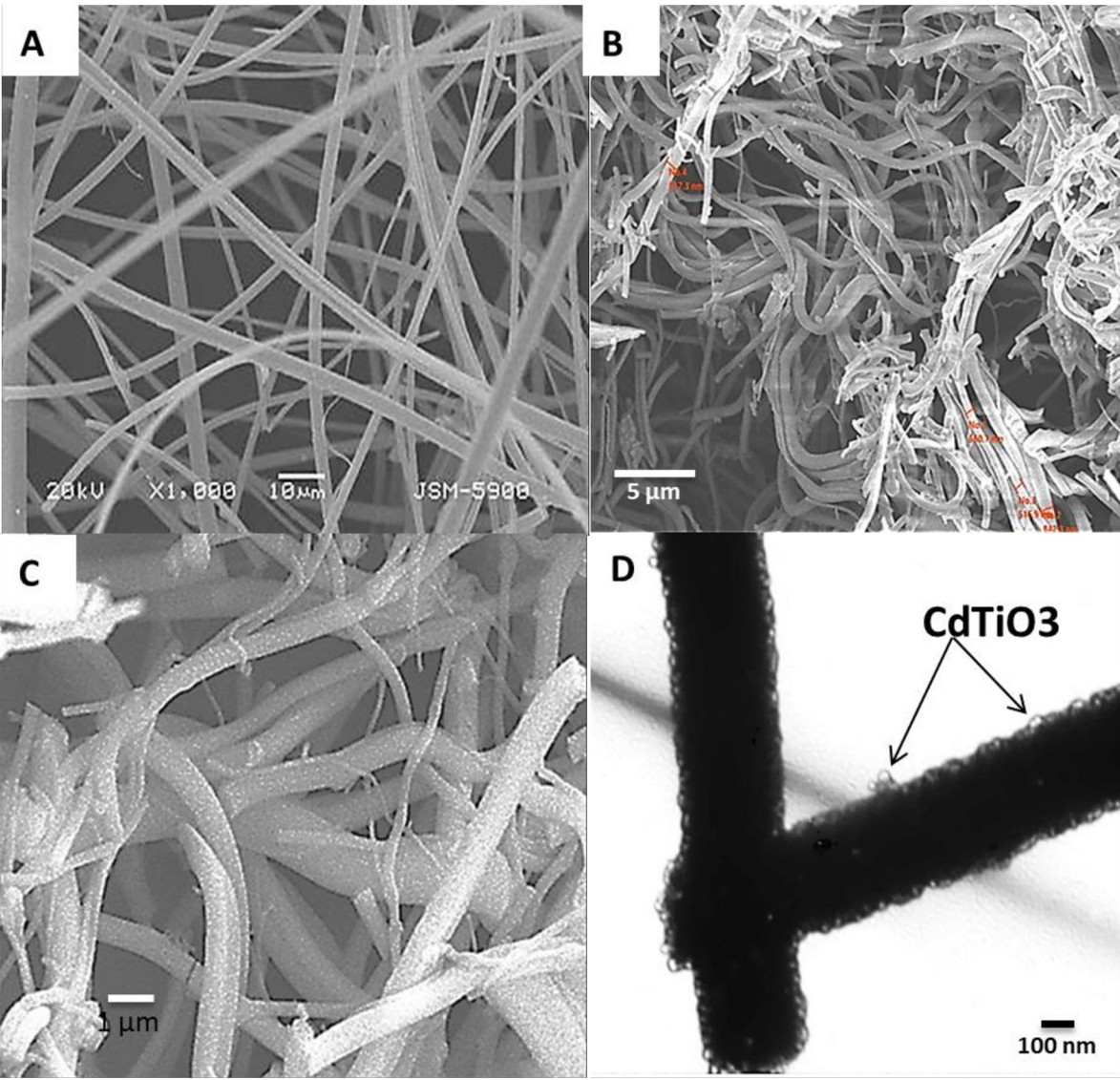

**Fig 2.** SEM images for pristine; (A) and 0.5 wt% Cd-doped TiO₂; (B) produced nanofibers after the calcination process. (C) magnification of 0.5 wt% Cd-doped TiO₂ nanofiber indicating the corrugated/speckled surface of the nanofiber. (D) TEM image of Cd-doped TiO₂ nanofiber indicating the attach of CdTiO₃ over the surface of TiO₂ nanofiber.

and the composition of the utilized sol–gel were properly selected. In other words, as a result of the polycondensation behaviors of all utilized precursors, the calcination process had no effect upon the nanomaterials' morphology.

It is notable that the addition of cadmium acetate did not distinctly affect the general morphology of the obtained nanofibers. The nanofiber sample chosen for SEM imaging is that having a composition of 0.5 wt% Cd-doped TiO$_2$ nanofibers. As observed in Fig 2B and 2C, cadmium incorporation results in the breaking of nanofibers. The produced nanofibers length decrease by increasing the cadmium content in the initial electrospinning solution. Typically results in evolving CO$_2$ and other exhausting gases leaving behind holes on the surface, potentially explaining the nanofiber breakdown. Comparing Fig 2B to 2A, the addition of cadmium can be seen to result in the creation of rough surface nanofibers. By contrast, the cadmium-free nanofibers have a smooth surface. A high surface area is a preferable characteristic for a photocatalyst, as it leads to enhanced photon absorption and consequently increases the catalytic activity. Fig 2D shows the TEM image of the Cd-doped TiO$_2$ nanofiber; the appearance of transparent bubble-like nanoparticles attached to the surface of the TiO$_2$ nanofiber is notable. It should be mentioned that the utilized calcination process for the prepared electrospun Ti (Iso)/CdAc)/PVP nanofibers resulted in producing CdO-TiO$_2$ (CdTiO$_3$) NPs-attaching a TiO$_2$ matrix [17, 39, 40]. As CdTiO$_3$ nanoparticles exhibit several catalytic, optical, and electrical properties, their presence over the surface of the nanofiber can enhance the TiO$_2$ nanofiber's photocatalytic performance by functioning as electron trapping sites for the hydrogen evolution reaction as it will be discussed below.

Fig 3 displays the TEM and the EDX results for the Cd-doped TiO$_2$ nanoparticles. The CdTiO$_3$ phase clearly appears in the TiO$_2$ matrix. As shown in the figure, CdTiO$_3$ has a different structure than that of TiO$_2$ (Fig 3B), which confirms the formation a Cd-doped TiO$_2$ nanostructure. The EDX result (Fig 3C) confirms the presence of Cd in the TiO$_2$ matrix. It is worth mentioning that EDX analysis has been conducted for 0.5% sample. The numerical data summarized in the inset table supports the aforementioned calculation hypotheses. Therefore, it can be concluded that, during the calcination process, the used polymer has been completely eliminated and the metallic precursors have been changed into the oxide forms without considerable losses in the metals content.

X-ray diffraction analysis is a typical technique for determining the composition of crystalline materials. Basically, brookite, anatase, and rutile are common TiO$_2$ phases from a crystal structure point of view; the latter two are abundantly found. Fig 4 represents the impact of cadmium doping on the crystal structure of the produced nanofibers. Cd-free electrospun nanofibers consist entirely of the anatase phase; the diffraction peaks appear at 2θ values of 25.09°, 37.65°, 38.44°, 47.89°, 53.89°, 55.07°, 62.40°, 68.70°, 70.04°, and 75.00° and correspond to the (101), (004), (112), (200), (105), (211), (204), (220), (220), and (215) crystal planes, respectively. This suggests the formation of pure anatase TiO$_2$ according to the XRD database (JCPDS card no 21–1272). One peak refers to the rutile phase at a 2θ value of 27.4° detected in the 0% Cd nanofiber sample. As can be seen in Fig 4, CdAc addition enhanced the rutile phase formation, which compared to anatase, has a greater tendency for visible light absorption [21, 22]. The tetragonal rutile phase (JCPDS 21–1276) at 2θ values of 27.4°, 36.1°, 39.2°, 41.2°, 44.1°, 54.3°, 56.6°, 62.7°, 64°, 69°, and 69.8° corresponds to the (110), (101), (200), (111), (210), (211), (220), (002), (310), (301), and (112) crystal planes, respectively. However, as shown, it can be concluded from the obtained data that 0.5% Cd nanofibers have the minimum amount of anatase; other combinations contain high amount. The reason for this may be that the addition of the optimum content of foreign dopant atoms into the lattice weakens its structure, thereby decreasing crystallinity [41]. The rutile (110) peaks shift towards lower angle for the samples

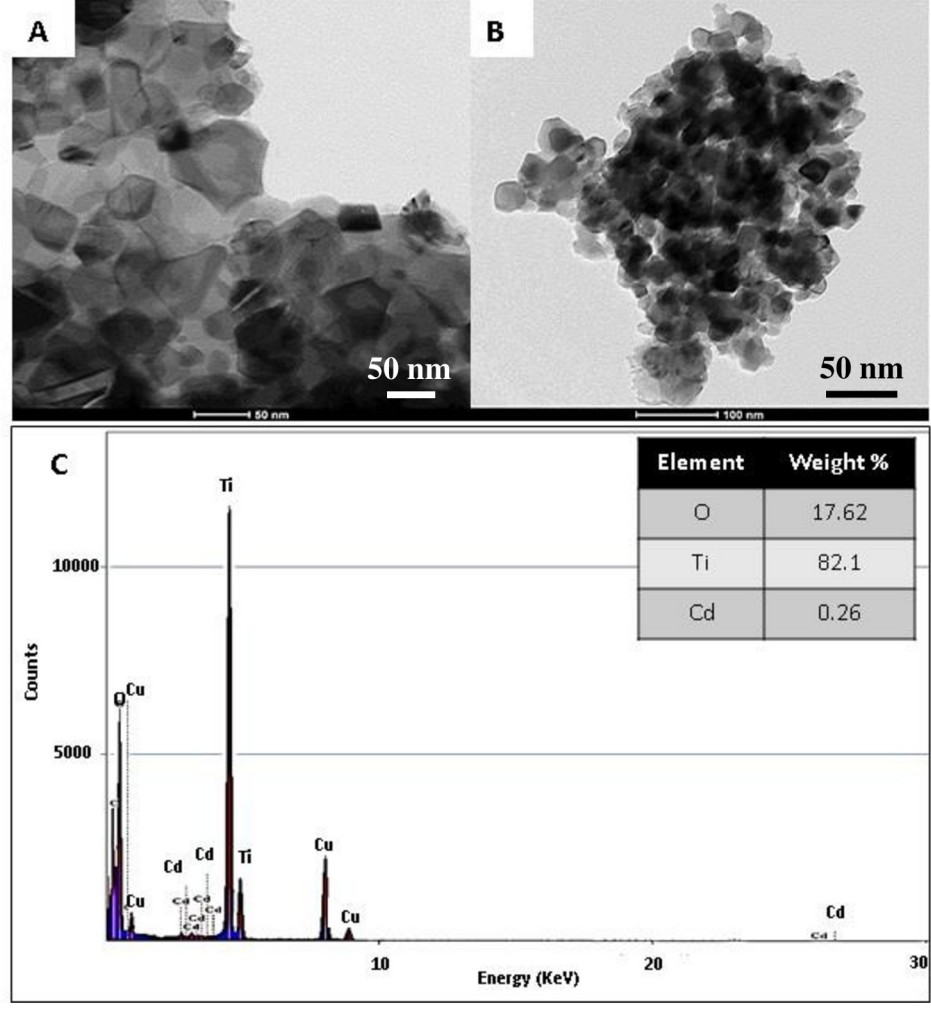

**Fig 3.** TEM images of pristine; (A) and CdO-doped; (B) $TiO_2$ nanoparticles. (C) The EDX image of $CdTiO_3$ doped $TiO_2$ nanoparticle obtained after atmospheric calcination at 700˚C.

with the addition of cadmium in addition to cadmium smaller radii compared to radius of $Ti^{+4}$ indicating the incorporation of cadmium metal ions in the $TiO_2$ matrix.

Due of the phase transition from the tetragonal (ilmenite) structure to the orthorhombic structure, $CdTiO_3$ crystallizes in both the ilmenite and perovskite structures and exhibits displacive-type ferroelectrics. In the perovskite structure, the $TiO_6$ octahedra are corner shared with the $Cd^{2+}$ ion having a 12-fold coordination. On the other hand, in the ilmenite structure, the $TiO_6$ octahedra are edge shared with the $Cd^{2+}$ ion possessing 6-fold-coordination [28]. At normal temperature, this semiconductor is existed in the form of tetragonal phase; however, to transform to the octahedral structure thermal treatment at elevated temperature is required; usually more than 1000 ˚C [42]. Diffraction peaks indicating formation of tetragonal (ilmenite) $CdTiO_3$ were observed at 31.1˚, 34.2˚, 46.9˚, and 59.3˚, corresponding to the (101), (104), (110), (024), and (214) crystal planes, respectively (JCPDS card no. 29–0277). The XRD result for Cd-doped $TiO_2$ nanoparticles shows a similar pattern to the nanofibers, as the XRD result does not depend upon the nanostructure's morphology; Fig 4B displays the nanoparticles prepared from 0.5 wt.% Cd solution.

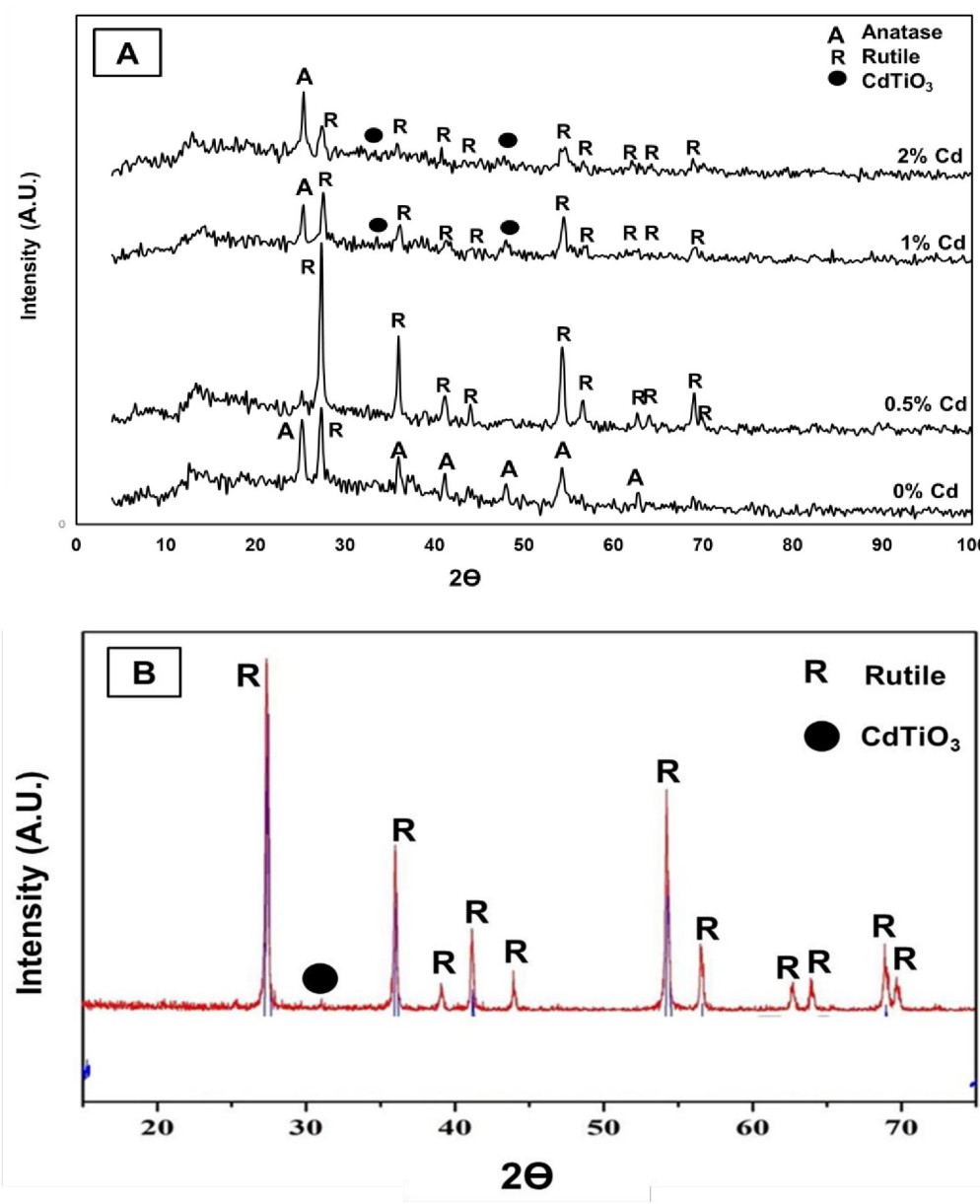

**Fig 4. The XRD patterns for pristine and Cd-doped TiO₂ nanofibers; (A) and Cd-doped TiO₂ nanoparticles (0.5 wt.% Cd sample); (B).**

## 3.2 Photocatalytic activity investigation

**3.2.1 Effect of nanomorphology and composition.** Fig 5 represents the effect of cadmium content upon the amount of hydrogen evolved under visible light irradiation using the prepared Cd-doped nanoparticles and nanofibers. As shown, the hydrogen production rates for the 2 wt.% Cd samples were 27 and 250 ml H₂/g$_{cat.}$ for nanoparticles and nanofibers, respectively. The nanofibrous morphology distinctly enhanced the photocatalytic activity. As observed in the figure, the rate of hydrogen production was increased greatly using nanofibers compared to nanoparticles at the same composition. The structure that provides one

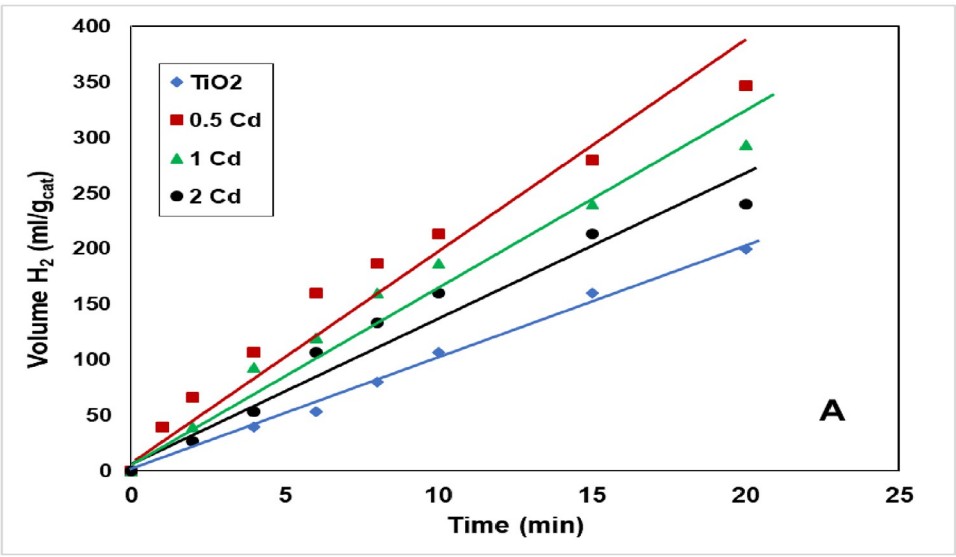

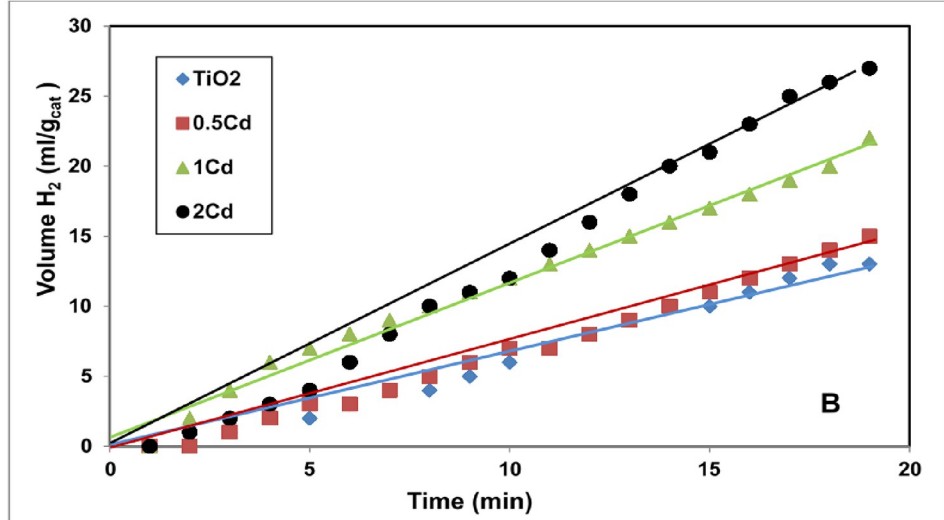

**Fig 5. Effect of Cd content on H₂ evolution rate in case of utilizing Cd-doped nanofibers; (A) and nanoparticles; (B) as photocatalyst.**

dimension for electron motion may be the reason for the considerable enhancement in the nanofibers' photocatalytic activity; however, in the nanoparticles with 0D structure, full electron confinement takes place, which favors the e⁻/h recombination process [43]. Moreover, the results show the effect of cadmium doping upon the photocatalytic activity. As shown in Fig 5, a remarkable enhancement in the photocatalytic activity of TiO₂ was detected after the addition of small amounts of cadmium. In fact, cadmium oxide particles act not only as visible light absorption centers with an absorption coefficient dependent on Cd density, but also as recombination sites between photogenerated electrons and holes. As aforementioned, doping by proper nanoparticles can yield to enlarge the Schottky barrier which increases electron capture which distinctly suppresses the recombination of e⁻/h pairs [20]. The observed decrease in the photocatalytic activity of the nanofibers for Cd concentrations above 0.5 wt% is related to

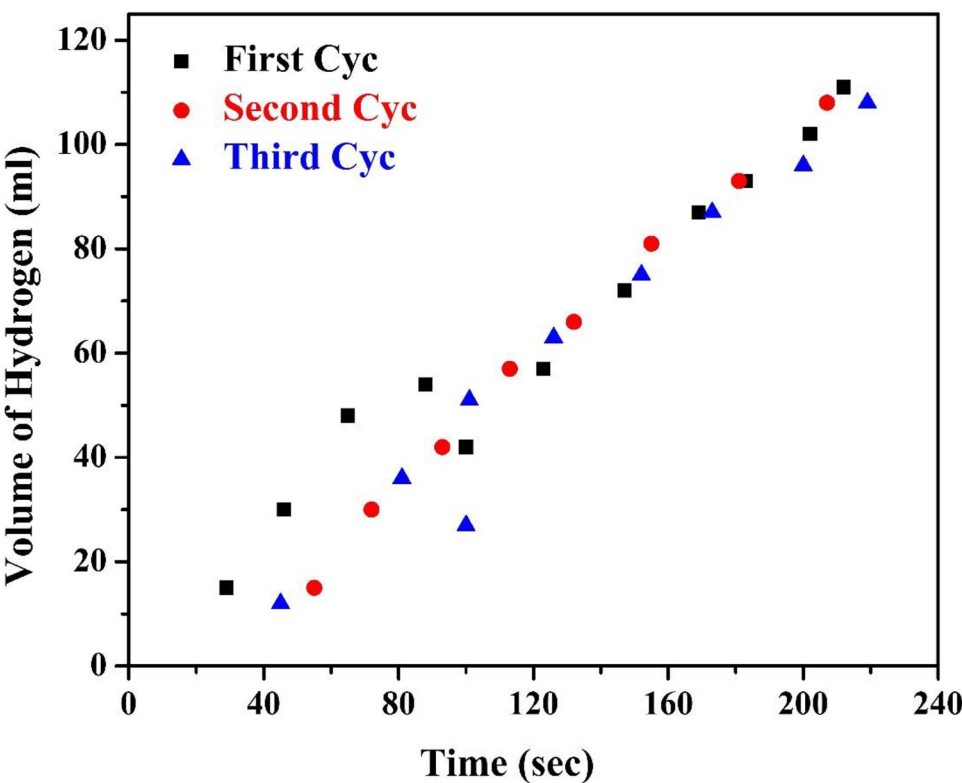

**Fig 6. Effect of multiple use of the introduced 0.5% Cd nanofiber.**

the decrease in rutile phase intensity, as confirmed by XRD results (Fig 4). Reusability is an important feature for the photocatalysts. Fig 6 demonstrates the obtained results after using the 0.5% Cd nanofiber for three successive cycles. As Fig 6 manifested, the nanofiber exhibited sustainable photocatalytic performance even after three cycles without significant loss of H$_2$ generation rate. These results confirm that the electrospun product was stable and reusable. Furthermore, such nanofibers could be effortlessly separated from the solution by facile sedimentation as the mixing stopped benefiting from the instinct feature of the large aspect ratio.

**3.2.2 Photosplitting efficacy.** Basically, the photocatalytic activity is intensely related to the optical properties of the light source used in the experiments, like the light intensity and irradiation area. Hence, the catalyst activities cannot be compared with each other if the reaction conditions are different. Therefore, determination of an apparent quantum yield (AQY), which rules the effect of light source out, is essential. Accordingly, it is necessary to determine the (AQY) of the present system and compare with other published works. To judge the feasibility of a proposed photocatalyst, the quantum efficiency of H$_2$ evolution is calculated as follows:

$$\eta = \frac{N_e}{N_v} \tag{2}$$

where $\eta$ is the energy conversion efficiency, $N_e$ is the number of electrons involved in the hydrogen evolution reaction, and $N_v$ is the number of incident photons in the reaction area. $N_e$ can be calculated from the rate of hydrogen evolution (from Fig 5, production rate = 0.63 mmol H$_2$ g$^{-1}_{\text{catalyst}}$min$^{-1}$ for Cd-doped TiO$_2$ nanoparticles). Through those rates, the actual

number of electrons disappearing due to the hydrogen evolution reaction can be obtained. The number of incident photons ($N_v$) was determined as $6.51\times10^{20}$ photons/s.m$^2$ according to literature [44]. This calculation indicates that the prepared catalyst has achieved a 20% conversion of photons to electrons, showing that the Cd-doped TiO$_2$ nanostructure photocatalyst is capable of increasing the absorbance of photons and prolonging the lifetime for e-/h pairs. Other researchers used another practical standard to calculate the efficiency of the photocatalystic water splitting; solar-to-hydrogen (STH) efficiency. It can be calculated using the following formula:

$$STH = \frac{Energy\ released\ from\ H_2\ gas}{Incident\ solar\ energy} = \frac{r_{H_2}\Delta G}{I_s A_r} \tag{3}$$

where $r_{H_2}$ is the hydrogen production rate in mol/s, $\Delta G$ is the Gibbs free energy associated with hydrogen gas in J/mol, $I_s$ is the solar energy flux in (W/cm$^2$), and $A_r$ is the surface area of the photocatalytic reactor. Using the STH formula, the STH efficiency of the prepared Cd-doped TiO$_2$ nanoparticles is calculated to be 0.64% and 9% for Cd-doped TiO$_2$ nanoparticle and Cd-doped TiO$_2$ nanofiber, respectively. Table 1 shows the rate of hydrogen evolution of this study compared with other works. It is important to note that comparison of the rate of hydrogen evolution should be done for experiments that are conducted using same light source and same sacrificing agent. However, it is hard to fulfill that condition to compare our results with other works. For nanostructures containing TiO$_2$, it is apparent that the moles of hydrogen produced by the Cd–doped TiO$_2$ nanofibers prepared in this study is relatively higher than that obtained previously by other scholars (Table 1); the exception is Mahmoud et al. [17], who indicated that a Cd-doped TiO$_2$ nanotube can achieve 24 mmol H$_2$/g$_{cat.}$ min using methanol as a scavenger agent. This reveals that the Cd-doped TiO$_2$ nanofiber may act as an effective photocatalyst candidate for the photon-induced water-splitting reaction. However, the stability and recyclability of this substance must be considered in greater detail before nominating it as a viable photocatalyst.

Table 1. A comparison of the hydrogen evolution rate for different nanocatalysts.

| Photocatalyst | Light source | Scavenger agent | H$_2$ production (mmol H$_2$/g$_{cat.}$ Min) | Ref./year |
|---|---|---|---|---|
| Pt/ TiO$_2$ nanosheet | Xenon Arc lamb 350 W | Ethanol | 0.0056 | [45]/2010 |
| Graphene modified TiO$_2$ nanoparticles | Xenon Arc lamb 350 W | Methanol | 0.0123 | [46]/2011 |
| TiO$_2$ nanoparticles | Xenon lamb 150 W | Methanol | 0.1 | [47]/2014 |
| (Pt/HS-TiO$_2$) | Mercury lamb 400 W | Methanol | 0.017 | [48]/2016 |
| Pt-doped TiO$_2$–ZnO | Mercury lamb 400 W | Methanol | 0.0034 | [49]/2017 |
| Pt-TiO$_2$ particles | Mercury lamb 450 W | Methanol | 0.444 | [53]/2005 |
| Cd-doped TiO$_2$ nanotube | Mercury lamb 1000 W | Methanol | 24 | [17]/2018 |
| CdS/TiO$_2$ mesoporous core-shell | Sunlight | Na$_2$S/ Na$_2$SO$_3$ | 1.13 | [18]/2018 |
| Ni/TiO$_2$ nanotube | Solar simulator Xenon lamb | - | 0.433 | [19]/2019 |
| Ni/GO-TiO$_2$ nanoparticles | Sunlight | Methanol | 3 | [50]/2019 |
| Ag-TiO2 NF | Mercury lamb 1000 W | Na$_2$S/ Na$_2$SO$_3$ | 2 | [51]/2020 |
| NiCo$_2$S$_4$/CdO@CC | Sunlight | - | 0.00125 | [52]/2020 |
| Cd-doped TiO$_2$ nanoparticles | Mercury lamb 1000 W | Na$_2$S/ Na$_2$SO$_3$ | 0.7 | This study |
| Cd-doped TiO$_2$ nanofibers | | | 16.5 | |

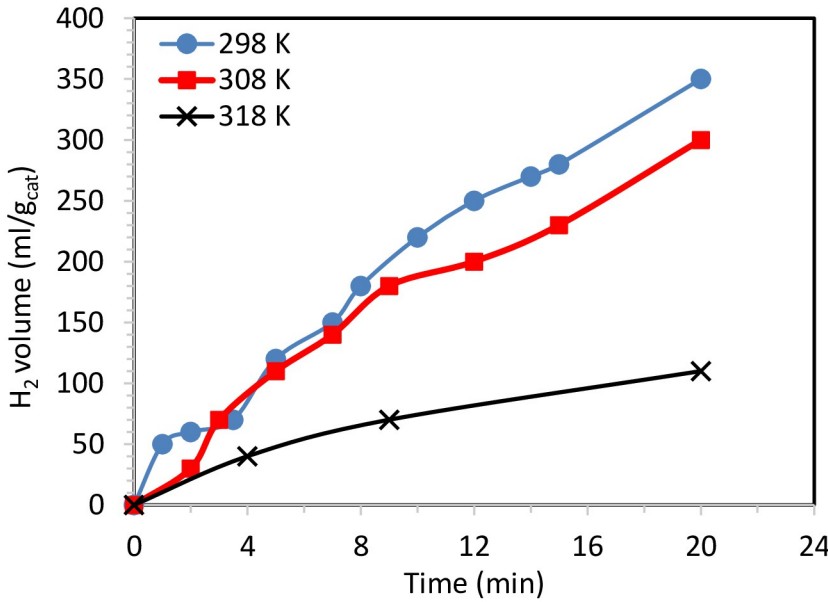

**Fig 7. Effect of temperature on hydrogen production from water using (0.05 g) of 0.5 wt % CdTiO$_3$-TiO$_2$ nanofiber.**

### 3.3 Influence of reaction temperature

The kinetic energy strongly depends on the temperature of the system, so higher temperature leads to a higher average molecular kinetic energy and more collisions per unit time. Therefore, in most chemical reactions, the temperature has a positive effect on the reaction rates. However, in hetero-catalytic reactions, increasing the reactants kinetic energy leads to a difficult catching of the reactants on the surface of the used solid catalyst which can be translated into negative impact of increasing the temperature on the reaction rate.

Fig 7 shows water photo splitting results using the proposed catalyst at temperatures of 298, 308, and 318 K. The results indicate that the hydrogen production rate decreases with increasing reaction temperature using Cd-doped TiO$_2$ nanofibers (0.5 wt% sample) as a photocatalyst. This behavior may be because the high kinetic energy might move the reactant molecules away from the active zones [21]. In addition, other researchers have noted that a surface plasmon is remarkably observed at low temperatures [53]. Moreover, increasing the temperature can increase the possibility of recombination between the charge holders, which consequently decreases the semiconductor's photoactivity. Therefore, we theoretically project that increasing the temperature is not preferred in the water-splitting reaction. This hypothesis has been verified experimentally (Fig 7). Numerically, the hydrogen production rates were 350, 300, and 100 ml/g$_{cat}$ at reaction temperatures of 298, 308, and 318 K, respectively. This finding confirms that the water-splitting reaction over the proposed catalyst surface does not follow the Arrhenius equation. Obtaining a high yield at low temperatures is an economically preferred characteristic in industrial settings.

### 3.4 Photocatalysis mechanism

Thermodynamically, in order to perform the redox reaction, the photocatalyst must have a conduction band with a potential level that is more negative than the redox potential of the unoccupied lowest molecular orbital of the photocatalyst-acceptor part. To trigger the

oxidation reaction, the valence band of the photocatalyst must have a potential level that is more positive than the reduction potential of the highest occupied molecular orbital in the photocatalyst-donor part. This mainly governs hydrogen and oxygen production over the surface of the photocatalyst. In other words, to start the $H^+/ H_2$ and $O_2/ H_2O$ reactions, both the conduction and valence band potentials of the photocatalyst must be wider than the hydrogen and oxygen production levels ($H^+/H_2$ (−0.41 V vs normal hydrogen electrode (NHE) at pH 7, $O_2/H_2O$ (+0.82 V vs NHE at pH 7) [54].

In practice, electron transfer is not as straightforward as it can be predicted. Previous research showed that electron transfer between semiconductors and aqueous redox species occurs only at the semiconductor/electrolyte interface, where two orbitals, one belonging to the semiconductor and the other to the aqueous species, have a comparable energy [55]. Furthermore, the energy difference between the energy levels (i.e. conduction band (CB) and lowest-unoccupied molecular orbital (LUMO) of the acceptor, and highest-occupied molecular orbital (HOMO) and VB of the donor) should be minimal. Sphalerite (ZnS), for example, has a strong conduction band (-3.46 eV against. absolute vacuum scale (AVS), -1.04 eV vs. NHE), however the large gap prevents it from catalyzing hydrogen synthesis photochemical reactions from water splitting [56]. In other words, while increasing the driving force (i.e., the energy difference between the electron donor and acceptor levels) is predicted to improve the rate of electron transfer processes, a high energy difference results in a sluggish electron transfer; this phenomenon is known as "*inverted region effect*" [57].

Numerically, the band gap and conduction band energies for $TiO_2$ are 3.2 and -0.29 eV vs. NHE (3.2 and -4.21 eV vs. AVS), respectively. While several reports have estimated the band gap of pure $CdTiO_3$ nanoparticles, the obtained value was around 2.9 eV [28, 58]. To properly understand the water splitting mechanism using the proposed catalyst, the conduction and valance bands of incorporated nanoparticles have to be determined. The required bands could be estimated from this empirical equations:

$$E_{CB} = \aleph - E^e - 0.5E_g \tag{4}$$

$$E_{VB} = E_{CB} + E_g \tag{5}$$

Where $E_{CB}$ and $E_{VB}$ are the conduction and valance potentials, respectively, $E^e$ is the energy free electrons vs. hydrogen (ca. 4.5 eV) [59]. The absolute electronegativity ($\chi$) of the semiconductor can be estimated from the following equation [60]:

$$\aleph = \left[ x(Ti)^a x(Cd)^b x(O)^c \right]^{1/(a+b+c)} \tag{6}$$

In which *a*, *b* and *c* are the number of atoms in the compound, *x(Ti)*, *x(Cd)* and *x(O)* are the absolute electronegativity of titanium, cadmium and oxygen elements; 3.45, 4.33 and 7.45 eV, respectively. Accordingly, the estimated values for the conduction and valance band energies were -0.22 and 2.68 eV, respectively.

Generally, the main gained benefit from doping a semiconductor by another one is bringing down the electron/hole pair recombination of the host semiconductor. According to the relative positions of the conduction and valance band energies, the dopant semiconductor can work as an electron sink or a hole sink or for both. Therefore, semiconductors heterojunctions can be classified into three types; straddling, staggered and broken band gap junctions. In the first type (straddling band gap), compared to the host semiconductor, the conduction band of the dopant is more negative while the valance is band is more positive. Consequently, the excited electrons and the formed holes jump to the dopant bands so the dopant acts as a sink for both of electrons and holes. Accordingly, the two redox reactions take on the surface of the

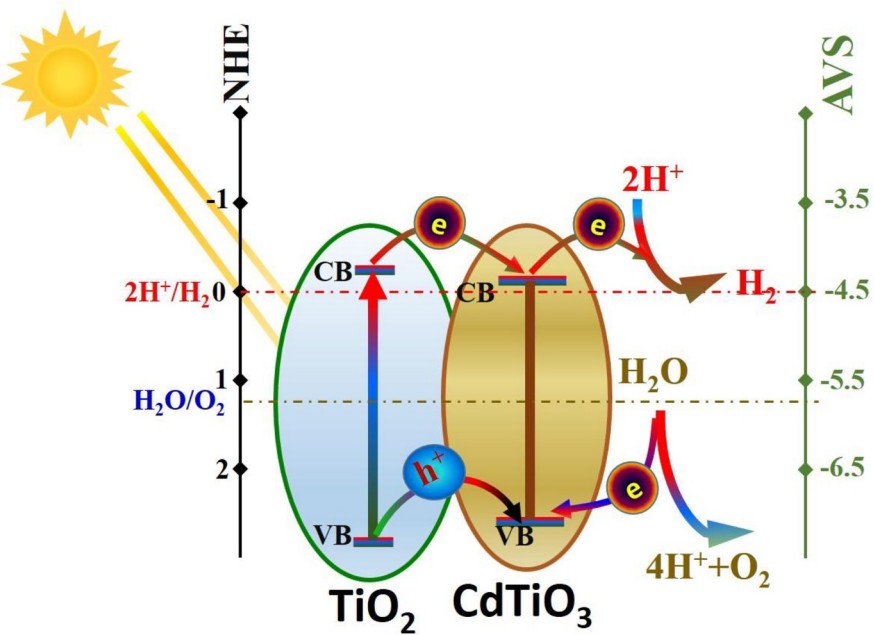

**Fig 8. The proposed mechanism for electron transfer and H$_2$ evolution on Cd-TiO$_2$ nanofibers.**

dopant semiconductor. For the second type, the dopant works as electron sink because both of the conduction and valance bands of the dopant are more negative than of that the host semiconductor. For the last type, there is a big difference between the energy bands. Typically, the valance band of the dopant semiconductor is more negative than the conduction band of the host semiconductor [34]. Fig 8 displays a schematic diagram for the location of titanium oxide and CdTiO$_3$ band energies. As shown in the diagram, it can be concluded that the proposed CdTiO$_3$ NPs-incorporated TiO$_2$ nanostructure belongs to "type 1" heterojunction semiconductor.

In water photo splitting, the scavenger is exploited to capture the holes which results in distinct depression for the electrons/holes recombination. In this study, Cd-incorporation is also proposed to suppress the electrons/holes recombination. To properly examine the success of the study target, water photo splitting process was examined without adding the scavenger; Fig 9. As shown in the figure, good hydrogen gas production rate was obtained from a suspension of Cd-incorporated TiO$_2$ nanofibers (0.5% Cd sample)/water (no Na$_2$S/Na$_2$SO$_3$ mixture was added) which indicates good termination of the electrons/holes recombination process. Therefore, it can be confidently claimed that using Cd incorporation results in producing an effective photocatalyst can be exploited in water photo splitting without a need of scavenger addition which is a highly recommended from the economical point of view.

In summary, the presence of CdTiO$_3$ with the TiO$_2$ nanostructure can create a type I heterojunction, which includes a spatial separation of electrons and holes, yielding higher reduction and oxidation potentials. Accordingly, this proposed photocatalyst can be effectively utilized to perform water splitting process under the visible light radiation without using a scavenger.

## 4. Conclusions

Cd-doped TiO$_2$ nanofibers, as a stable and nonprecious catalyst for water-splitting reactions, can be produced by simple, high yield and low cost process; electrospinning of a sol–gel

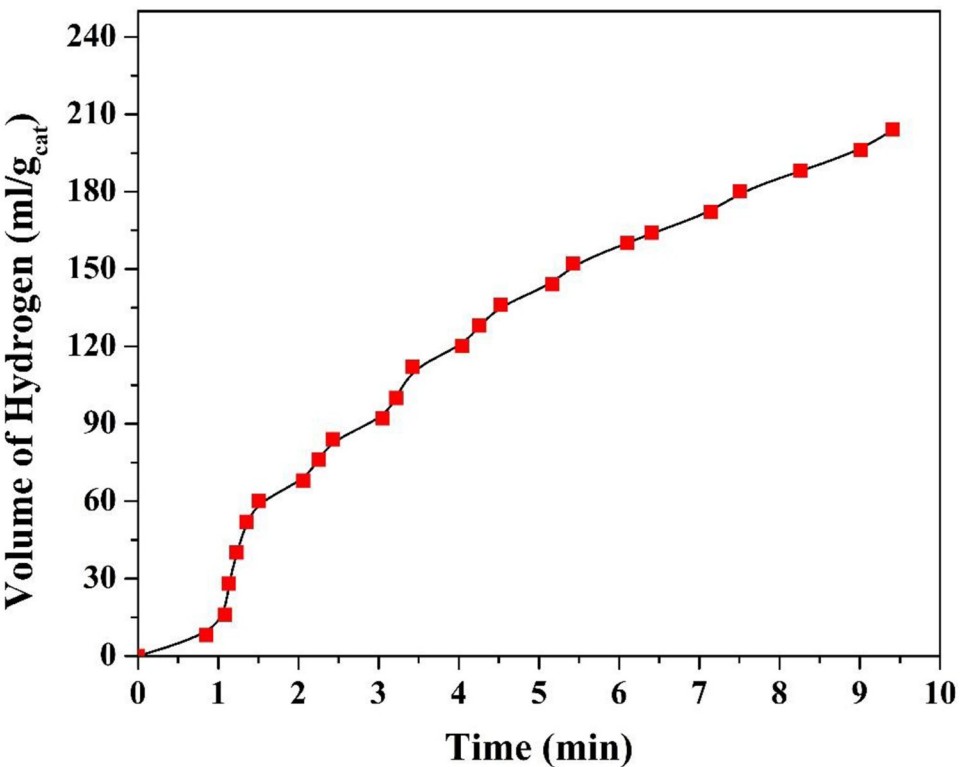

**Fig 9. Hydrogen production rate using 0.5% Cd nanofiber sample at 25 ˚C without using a scavenger.**

composed of titanium isopropoxide, cadmium acetate and polyvinylpyrrolidone followed by heat treatment at high temperature. The high polycondensation tendency of the used metals precursors resulted in maintaining the nanofibrous morphology during the calcination process. Both of cadmium oxide content and nanomorphology has a highly considerable impact on the photocatalytic activity of the proposed composite. The optimum cadmium content depends on the produced nanostructure morphology. Typically, the maximum hydrogen production rate can be obtained with CdTiO$_3$-doped TiO$_2$ nanofibers and nanoparticles containing 0.5 and 2 wt.% dopant, respectively. Doping of TiO$_2$ by Cd shows a good influence on increasing the electrons/holes lifetime. Accordingly, it presented good photocatalytic activity for water splitting which is translated as a distinct increase in the hydrogen evolution rate compared to that of undoped titanium oxide nanoparticles and nanofibers. Moreover, the proposed catalyst can be utilized for water splitting in a scavenger-free normal water.

## Author Contributions

**Data curation:** Mohamed S. Mahmoud.

**Formal analysis:** Nasser A. M. Barakat.

**Funding acquisition:** Hak Yong Kim.

**Methodology:** Nehal A. Erfan, Mohamed S. Mahmoud.

**Supervision:** Nasser A. M. Barakat.

**Writing – original draft:** Nehal A. Erfan, Nasser A. M. Barakat.

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
