## [Decision Letter · Decision Letter 0]

22 Aug 2022

PONE-D-22-20788CdTiO3 NPs-incorporated TiO2 nanostructures as effective photocatalyst for possible scavenger-free water photo splitting process under visible light radiationPLOS ONE

Dear Dr. Barakat,

Thank you for submitting your manuscript to PLOS ONE. After careful consideration, we feel that it has merit but does not fully meet PLOS ONE’s publication criteria as it currently stands. Therefore, we invite you to submit a revised version of the manuscript that addresses the points raised during the review process.

We look forward to receiving your revised manuscript.

Kind regards,

Satya Pal Nehra, PhD

Academic Editor

PLOS ONE

Journal Requirements:

Additional Editor Comments:

Reviewers have now commented on your paper. You will see that there are a number of issues that need to be addressed before the paper can be accepted for publication by PLOS ONE. I ask that you give the comments raised by the referees your careful consideration and that you submit a revised version of your manuscript as well as an itemized reply to each of the reviewers' comments.

I am looking forward to receiving you revised manuscript!

Yours sincerely,

Dr. Satya Pal Nehra

Academic Editor

PLOS ONE

Reviewers' comments:

Reviewer's Responses to Questions

**Comments to the Author**

1. Is the manuscript technically sound, and do the data support the conclusions?

Reviewer #1: Yes

Reviewer #2: Yes

2. Has the statistical analysis been performed appropriately and rigorously? 

Reviewer #1: Yes

Reviewer #2: Yes

3. Have the authors made all data underlying the findings in their manuscript fully available?

Reviewer #1: Yes

Reviewer #2: Yes

4. Is the manuscript presented in an intelligible fashion and written in standard English?

Reviewer #1: Yes

Reviewer #2: Yes

5. Review Comments to the Author

Reviewer #1: Erfan and co-author created a very interesting and valuable study dealing with the preparation of Cd(II) dopped TiO2 catalysts in different ratios. The photocatalytic potentials of prepared materials were investigated in the hydrogen production by water splitting reaction. The presented manuscript can be considered for publication in PLOS ONE after major revision, addressing of following points:

Introduction: ... or the use of a photocatalyst for water splitting [4, 5]. ... List the following review dealing with hydrogen production using GCN: R. Sharma, M. Almáši, S.P. Nehra, V.S. Rao, A. Sharma, I.P. Jain, Photocatalytic hydrogen production using graphitic carbon nitride (GCN): A precise review, Renew. Sust. Energ. Rev. (2022) in press, https://doi.org/10.1016/j.rser.2022.112776

Materials - It would be appropriate to unify information related to chemicals: (abbreviation, summary formula, Mw, purity). The chemical composition of cadmium acetate is not correct, 2 moles of water are missing.

The volume of released hydrogen was not determined directly, e.g. GC, but indirectly by monitoring the change in water volume. During the experiments, a 1000 W lamp was used, which produced a lot of heat. How do the authors prevent water evaporation? Because the aforementioned evaporation of water introduces measurement error.

The authors prepared three doped TiO2 materials with Cd(II), where cadmium acetate was added during the sol-gel synthesis in different ratios of 0.5; 1 and 2. But what is the real content of Cd(II) in the prepared solid samples? (possible solution: EDX, Rietveld analysis form PXRD, mineralization of materials in aqua regia and further AAS or ICP-MS measurements).

From an environmental point of view, the authors should investigate the possible leaching of Cd(II) toxic ions into the water after water splitting reaction (e.g. ICP-MS or AAS). Moreover, the H2 evolution rate of Cd(II) doped samples decreases with time compared to pure TiO2, which also indicates possible leaching.

What is the effectiveness (recyclability) of the catalyst (e.g. material with 0.5 Cd(II) content) after multiple uses (for example, 5 cycles)?

Reviewer #2: The selected topic covers a sufficient range of general interest including at the same time novel scientific aspects. The experimental setup is well presented along with the interpretation of the obtained data. Diagrams and images are adequately attached and the literature included is spherical enough of what's been done so far. Further a few suggestions are from my side to make the manuscript complete:

(i) The title seems too long, it’s good to limit it in 10-13 words. An example of suggested title from my side is “CdTiO3-NPs incorporated TiO2 nanostructure photocatalyst for scavenger free water splitting under visible radiation”. If you feel it retain the sense of your article, consider it.

(ii) On Page 2., instead of writing “It is hypothesized that the formation of type I heterostructures between the TiO2 matrix and CdTiO3 nanoparticles….” Use “may be” to show your hypothesis.

(iii) In Introduction section on Page 3, non-metallic doping lacked some new research. Some related latest literature should be updated, such as RSER,2022,168:112776(https://doi.org/10.1016/j.rser.2022.112776); Chemosphere,2022,305:135467(https://doi.org/10.1016/j.chemosphere.2022.135467); IJHE,2020,45:23937 (https://doi.org/10.1016/j.ijhydene.2019.06.061).

(iv) Do not use e/h, rather put symbols also (e-/h).

Once the authors complete these few changes, I’ll suggest it’s acceptance for the publication.

6. PLOS authors have the option to publish the peer review history of their article (what does this mean?). If published, this will include your full peer review and any attached files.

Reviewer #1: No

Reviewer #2: **Yes: **Rishabh Sharma

---

## [Author Response · Author response to Decision Letter 0]

21 Sep 2022

Dear Dr. Satya Pal Nehra

Academic Editor

PLOS ONE

Thank you for your kind response about the manuscript [PONE-D-22-20788] titled:

 “CdTiO3 NPs-incorporated TiO2 nanostructures as effective photocatalyst for possible scavenger-free water photo splitting process under visible light radiation”

The referee's comments were helpful to strength the manuscript. We would like to inform you that we have modified the manuscript according to the newly given comments. 

To make it more easily, we have written the comments in bold phase followed by the responses in normal one. Moreover, in the revised manuscript, you can find the changes in the text in blue color. 

 We hope our responses cover all the comments. It will be our pleasure to respond about any more comments.

Thank you for your cooperation 

Sincerely yours 

Corresponding author

Nasser A. M. Barakat

Professor

Chemical engineering dep., Minia university, Egypt

Reviewer #1: 

Erfan and co-author created a very interesting and valuable study dealing with the preparation of Cd(II) dopped TiO2 catalysts in different ratios. The photocatalytic potentials of prepared materials were investigated in the hydrogen production by water splitting reaction. The presented manuscript can be considered for publication in PLOS ONE after major revision, addressing of following points:

1- Introduction: ... or the use of a photocatalyst for water splitting [4, 5]. ... List the following review dealing with hydrogen production using GCN: R. Sharma, M. Almáši, S.P. Nehra, V.S. Rao, A. Sharma, I.P. Jain, Photocatalytic hydrogen production using graphitic carbon nitride (GCN): A precise review, Renew. Sust. Energ. Rev. (2022) in press, https://doi.org/10.1016/j.rser.2022.112776

Response:

Thank you for this comment, adding this important review will enhance the manuscript.

The review has been added to the revised manuscript as reference 6

2. Materials - It would be appropriate to unify information related to chemicals: (abbreviation, summary formula, Mw, purity). The chemical composition of cadmium acetate is not correct; 2 moles of water are missing.

Response:

The reviewer is right; the chemicals information is unified in the updated manuscript version. Concerning cadmium acetate, it was out mistake, we have checked the used bottle and discovered that the reviewer is right; it associated with two water molecule so this mistake has been corrected in the revised manuscript. 

3-The volume of released hydrogen was not determined directly, e.g. GC, but indirectly by monitoring the change in water volume. During the experiments, a 1000 W lamp was used, which produced a lot of heat. How do the authors prevent water evaporation? Because the aforementioned evaporation of water introduces measurement error.

Response: As usual, the reviewer arises very valuable comment. 

It is noteworthy mentioning that to ensure the collected gas represents the water splitting reaction product gases and no interfere from water vapor especially the used lamp becomes hot during illuminating time, these precautions have been conducted. First, the lamp was cooling by a suction fan placed above the lamp so this used fan sucks the hot air around the reactor flask. Second, the collecting gas beaker was placed a way from the lamp so it was always cold. Third, the used water in the collector beaker was checked continuously during the reaction time to be sure it still cold. Fourth, after every experiment, the collect gas was left for relatively long time to allow any water vapor to condensate; no any change in the gas volume was observed which indicates the collected gas is free from any water vapor.

This paragraph was added in the revised manuscript. 

4- The authors prepared three doped TiO2 materials with Cd(II), where cadmium acetate was added during the sol-gel synthesis in different ratios of 0.5; 1 and 2. But what is the real content of Cd(II) in the prepared solid samples? (possible solution: EDX, Rietveld analysis form PXRD, mineralization of materials in aqua regia and further AAS or ICP-MS measurements).

Response: This comment came also from another mistake in the initial manuscript; we strongly appreciate the respected reviewer efforts. 

To prepare final nanofiber mats containing 0.5, 1, and 2 wt.% CdO with respect to titanium element, the corresponding mass of cadmium acetate is added. The calculations were based on assumptions that a complete elimination of the used polymer and full decompositions of titanium isopropoxide and cadmium acetate di-hydrate precursors into titanium oxide and cadmium oxide, respectively will carry out. The calcination process was done in air atmosphere. Moreover, to ensure performing complete oxidation reactions, suction pump was used to remove the gaseous products from the reaction atmosphere as well as to fill the tubular furnace by fresh air continuously. Accordingly, the claiming about complete elimination of the used polymer and converting the used metallic precursors to the oxide forms is acceptable. Considering that the melting points of titanium oxide and cadmium oxide are 1843 and 900 oC, respectively, no losses in the metal content are expected especially the calcination temperature was chosen to be 200 oC less than the melting point of the relatively volatile oxide (CdO).

This explanation has been added in the section ”2.2 Fabrication of TiO2 and Cd-doped TiO2 nanofibers and nanoparticles” in the revised manuscript. 

Moreover, EDX has been conducted (Fig. 3) in the revised manuscript with this explanation. 

“It is worth mentioning that EDX analysis has been conducted for 0.5 % sample. The numerical data summarized in the inset table supports the aforementioned calculation hypotheses. Therefore, it can be concluded that, during the calcination process, the used polymer has been completely eliminated and the metallic precursors have been changed into the oxide forms without considerable losses in the metals content” 

5- From an environmental point of view, the authors should investigate the possible leaching of Cd(II) toxic ions into the water after water splitting reaction (e.g. ICP-MS or AAS). Moreover, the H2 evolution rate of Cd(II) doped samples decreases with time compared to pure TiO2, which also indicates possible leaching.

Response: It is also good comment. Unfortunately, due to COVID-19 pandemic we could not do some analyses. However, the stability of proposed photocatalysts was investigated through checking the stability at different successive cycles. In the revised manuscript, Figure 6 was newly added. In this figure, 0.5 Cd nanofibers were used for three successive cycles. As shown in the figure, almost no change in the activity can be observed which denotes high stability of the proposed photocatalyst. 

 6- What is the effectiveness (recyclability) of the catalyst (e.g. material with 0.5 Cd(II) content) after multiple uses (for example, 5 cycles)?

Response:

To provide a proper response for the given comment, Fig. 6 has been updated to investigate the stability of 0.5% Cd nanofiber. The discussion has been updated as follow:

Reusability is an important feature for the photocatalysts. Figure 6 demonstrates the obtained results after using the 0.5 % Cd nanofiber for three successive cycles. As figure 6 manifested, the nanofiber exhibited sustainable photocatalytic performance even after three cycles without significant loss of H2 generation rate. These results confirm that the electrospun product was stable and reusable. Furthermore, such nanofibers could be effortlessly separated from the solution by facile sedimentation as the mixing stopped benefiting from the instinct feature of the large aspect ratio. 

Reviewer #2: 

The selected topic covers a sufficient range of general interest including at the same time novel scientific aspects. The experimental setup is well presented along with the interpretation of the obtained data. Diagrams and images are adequately attached and the literature included is spherical enough of what's been done so far. Further a few suggestions are from my side to make the manuscript complete:

We strongly appreciate the great efforts of the respected reviewer in evaluating the manuscript; the given comments were valuable to strength the manuscript. 

1- The title seems too long, it’s good to limit it in 10-13 words. An example of suggested title from my side is “CdTiO3-NPs incorporated TiO2 nanostructure photocatalyst for scavenger free water splitting under visible radiation”. If you feel it retain the sense of your article, consider it.

Response:

The suggested title is concise and meaningful so we changed the title in the revised manuscript accordingly.

2- On Page 2., instead of writing “It is hypothesized that the formation of type I heterostructures between the TiO2 matrix and CdTiO3 nanoparticles….” Use “may be” to show your hypothesis.

Response:

The text has been updated in the revised manuscript

3- In Introduction section on Page 3, non-metallic doping lacked some new research. Some related latest literature should be updated, such as RSER,2022,168:112776(https://doi.org/10.1016/j.rser.2022.112776); Chemosphere,2022,305:135467(https://doi.org/10.1016/j.chemosphere.2022.135467); IJHE,2020,45:23937 (https://doi.org/10.1016/j.ijhydene.2019.06.061).

Response:

It is believed that the suggested references strength the manuscript, so they were cited in the revised version 

4- Do not use e/h, rather put symbols also (e-/h).

Response:

The text has been updated in the revised version

---

## [Decision Letter · Decision Letter 1]

29 Sep 2022

CdTiO3-NPs Incorporated TiO2 Nanostructure Photocatalyst for Scavenger-free Water Splitting Under Visible Radiation

PONE-D-22-20788R1

Dear Dr. Nasser,

We’re pleased to inform you that your manuscript has been judged scientifically suitable for publication and will be formally accepted for publication once it meets all outstanding technical requirements.

Kind regards,

Satya Pal Nehra, PhD

Academic Editor

PLOS ONE

Additional Editor Comments (optional):

Authors have revised the manuscript. It is recommended for publication in PLOS ONE.

Reviewers' comments:

Reviewer's Responses to Questions

**Comments to the Author**

1. If the authors have adequately addressed your comments raised in a previous round of review and you feel that this manuscript is now acceptable for publication, you may indicate that here to bypass the “Comments to the Author” section, enter your conflict of interest statement in the “Confidential to Editor” section, and submit your "Accept" recommendation.

Reviewer #1: All comments have been addressed

Reviewer #2: All comments have been addressed

2. Is the manuscript technically sound, and do the data support the conclusions?

Reviewer #1: Yes

Reviewer #2: Yes

3. Has the statistical analysis been performed appropriately and rigorously? 

Reviewer #1: Yes

Reviewer #2: Yes

4. Have the authors made all data underlying the findings in their manuscript fully available?

Reviewer #1: Yes

Reviewer #2: Yes

5. Is the manuscript presented in an intelligible fashion and written in standard English?

Reviewer #1: Yes

Reviewer #2: Yes

6. Review Comments to the Author

Reviewer #1: All comments were incorporated and answered in the manuscript. I can only recommend accepting an article in PlosOne. In conclusion, I wish the authors many citations for the submitted manuscript.

Reviewer #2: (No Response)

7. PLOS authors have the option to publish the peer review history of their article (what does this mean?). If published, this will include your full peer review and any attached files.

Reviewer #1: **Yes: **Miroslav Almasi

Reviewer #2: No

---

## [Editor Report · Acceptance letter]

7 Oct 2022

PONE-D-22-20788R1 

CdTiO3-NPs Incorporated TiO2 Nanostructure Photocatalyst for Scavenger-free Water Splitting Under Visible Radiation 

Dear Dr. Barakat:

I'm pleased to inform you that your manuscript has been deemed suitable for publication in PLOS ONE. Congratulations! Your manuscript is now with our production department. 

Kind regards, 

on behalf of

Dr. Satya Pal Nehra 

Academic Editor

PLOS ONE